# Monitoring the Dynamics of Ephemeral Rivers from Space: An Example of the Kuiseb River in Namibia



Cassandra Normandin [1,*], Philippe Paillou [1], Sylvia Lopez [1], Eugene Marais [2] and Klaus Scipal [3]

1 UMR 5804, University of Bordeaux, 33615 Pessac, France
2 Gobabeb Namib Research Institute, Walvis Bay P.O. Box 953, Namibia
3 European Space Agency, ESRIN, 00044 Frascati, Italy
* Correspondence: cassandra.normandin@u-bordeaux.fr

**Abstract:** Ephemeral rivers are characterized by brief episodic flood events, which recharge subterraean alluvial aquifers that sustain humans, riparian vegetation, and wildlife in the hyper-arid Namib Desert. Yet we only have a poor understanding of the dynamics and feedback mechanisms in these hydrological systems as arid and semi-arid zones are typically poorly equipped with reliable in situ monitoring stations to provide necessary information. The main objective of our study is to show the potential of satellite data to monitor the dynamics of ephemeral rivers, such as the Kuiseb located in Namibia, since remotesensing offers the advantage of adapted spatial and temporal resolutions. For this study, multi-spectral imagery (Sentinel-2), Synthetic Aperture Radar (SAR, Sentinel-1), and SAR interferometry (Sentinel-1) data were used to produce Normalized Difference Vegetation Index (NDVI) maps, backscattering maps (as $\sigma_0$), and interferograms, respectively. These parameters provide information on the hydrologic and vegetation dynamics of the river. Strong variations in NDVI, $\sigma_0$, and interferograms are observed during March–April 2017 and June–July 2018 in a tributary of the Kuiseb in the central Namib Desert. In those years, rain events caused the reactivation of the tributary. However, during a major flood in 2021, when no rain occured, no variations in NDVI were detected in this tributary, unlike the $\sigma_0$ and interferogram anomalies after the flood. Thus, these variations cannot be explained by rains, which were non-existent during this period, but seem to be linked to the dynamics of the aquifer of the Kuiseb River, wherein floods recharge the alluvial aquifers and the rising water table levels produce a signal that is measurable by satellite radar sensors. All these results present a preliminary work that might be used by water resource managers to automate the processing and methods used to create an ephemeral river monitoring tool.

**Keywords:** fluvial aquifer hydrology; groundwater monitoring; multispectral imagery; radar backscattering; interferometry

## 1. Introduction

Arid zones are characterized by low rainfall, long periods of drought, and frequent water scarcity. These regions cover about 41% of the earth's land surface [1,2] and are typically in areas where there is a divergence of atmospheric circulation patterns at the surface, such as on the lee side of mountains chains or near cold oceanic regions. Aridity results from an imbalance between evapotranspiration and rainfall, thus reliable water resources in arid zones are scarce and important [3,4]. In addition to rainfall, fog and dew may also be an important potential source of water in arid areas for vegetation [5,6]. Despite arid zone vegetation being mainly herbaceous with few trees, it still contributes 40% of the global net primary productivity and plays a key role in carbon sequestration [2,7].

Worldwide, one in every three inhabitants lives in an arid zone. Most of the water resources that people depend on are concentrated along ephemeral rivers, which are characterized by surface water flows ranging from a few hours to a few days per year, with measurable flows occurring less than 10% of the year [3,8]. Water flowing in the beds

of these rivers usually takes place as flood events after rain episodes in their catchments. Such floods recharge groundwater aquifers by seepage of water into the alluvial substrate of the riverbed and on floodplains [3,8], which are a major source of freshwater in arid areas. Understanding floods and the associated hydrological dynamics is therefore crucial for determining sustainable yields in global water resources. Moreover, in the context of climate change, arid and semi-arid zones are likely to increase globally, and therefore the number of ephemeral rivers will also increase. Improved monitoring and understanding of the dynamics of ephemeral rivers and associated groundwater resources is therefore an essential component of adaptation to climate change.

Until recently, the monitoring of ephemeral rivers and the associated groundwater aquifers was mainly through in situ hydrological and geohydrological stations [3] to gather data and model their dynamics [9]. However, a major limitation for in situ data from arid zones is the low number of available monitoring stations and data gaps from inoperable stations due to the high cost of installation and maintenance of such stations. Adequate spatial coverage, which was adapted to the study of an ephemeral river system as a whole, and continuous data to identify key hydrological fluxes, are rarely available. Models, for their part, represent a complementary tool, but they require reliable in situ data for their calibration and validation.

Remote sensing from Earth observation satellites offers a low-cost opportunity to monitor ephemeral rivers and their associated groundwater resources, allows access to data over extended time periods, and can cover the specific spatial resolution that may be required. In the presented study here, we benefitted from ESA Earth observation missions—in particular the Sentinel series of the EU Copernicus program. The Sentinel satellites, currently consisting of six satellites in orbit, offer a rich multi-sensor Earth observation dataset. We tested an innovative multi-sensor approach, as a preliminary work, using both Sentinel-1 and Sentinel-2 data to evaluate the potential for monitoring the ephemeral Kuiseb River and the dynamics of its fluvial aquifers in Namibia. We extracted parameters such as flood extent, vegetative growth, and backscattering coefficient, which provide information on the subsurface properties and soil moisture, which were in archived time-series from at least three years of continuous acquisitions by the Sentinel satellites through Synthetic Aperture Radar (SAR) and multi-spectral sensors.

## 2. Ephemeral Rivers

### 2.1. General Information

Ephemeral and intermittent rivers, characterized by occasional, periodic, or recurrent water flows, represent about half of all rivers on Earth and are located in different climates: temperate, boreal, humid tropical, and arid [10–12]. Here, we only focus on ephemeral rivers, which are defined by having measurable flow in the riverbed for less than 10% of the year and with flood peaks that last from a few hours to a few days [3,8]. In recent years, the number of ephemeral rivers has increased worldwide as a consequence of the combined effects of climate change and land use changes, which have reduced discharge in historically more constant flowing rivers [11,13]. Reduced flows do not only have a significant effect on human populations and their livelihoods along river courses, but also have significant repercussions on biodiversity and associated ecosystems.

Ephemeral rivers first interested scientists in the fields of biodiversity and ecology, since they consist of a multitude of habitats (terrestrial, lotic, lentic) with a rich biodiversity of resilient species that are resistant to the extreme fluctuations between aquatic and terrestrial conditions [11,14]. Although there is a continued interest in the ecological understanding of ephemeral rivers, studies on hydrological aspects remain limited. Even though hydrologists started to pay more attention to ephemeral rivers and their flood intervals due to the importance of these rivers as the main source of water to humans and biodiversity in arid and semi-arid zones, they are hampered by data limitations [3]. Water arrives quickly in the main bed of the river during periods of flooding, and then infiltrates into the substrate at variable speeds, which drives the development of vegetation

and recharge of aquifers [3,10]. It is this dynamic between sudden floods, subsequent infiltration, and the response of vegetation and rising groundwater levels that is of interest to hydrologists.

The limited number of studies on the hydrological processes of ephemeral rivers are mainly due to inadequate and inappropriate tools for monitoring the brief and highly dynamic hydrological phenomena of these systems [12]. Indeed, most in situ gauging stations have been developed for and are located on permanent rivers, being very rare in arid and semi-arid areas. They are even more rarely combined with geohydrological gauges to monitor infiltration rates and fluctuations in groundwater levels. Fortunately, for about two decades, the increased sophistication and availability of remote sensing data have offered new tools for the study of hydrological phenomena in ephemeral rivers.

### 2.2. The Kuiseb River

The Kuiseb River is one of twelve major west-flowing ephemeral rivers in Namibia, traversing the central Namib Desert on the southwestern African coastal plain (Figure 1a,b). It is one of the longest ephemeral rivers in Namibia, with a length of 560 km and a drainage basin of 15,500 km$^2$ [15]. The Kuiseb River begins its course on the high plateau of the Khomas Hochland Mountains (1,500–2,000 m above sea level), cutting through the Great Escarpment [15] before crossing the Namib Desert to reach the Atlantic Ocean at Walvis Bay [3,16]. The river valley is largely incised into schists and quartzite bedrock, of which the erosion products are the source of an alluvial sand and gravel, which fill in its riverbed. The lower part of the Kuiseb River, approximately 150 km from the coast, below the Kuiseb Canyon, consists of a sandy channel bordered by riparian vegetation before widening at Rooibank, approximately 30 km from the ocean, into a terminal delta [8,16].

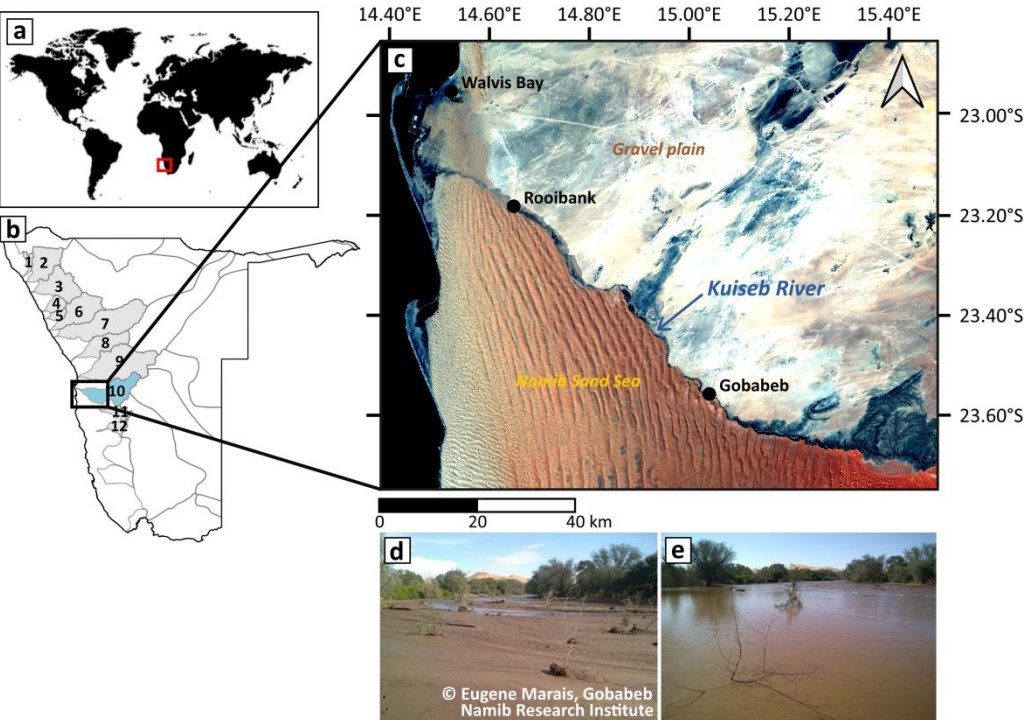

**Figure 1.** (**a**) World map and location of Namibia with a red box. (**b**) Locations of the 12 ephemeral rivers in Namibia, with the Kuiseb River highlighted in blue (number 10, from https://namibia. africageoportal.com/ (accessed on 4 April 2022)). (**c**) Sentinel-2 image of the Kuiseb River, (**d**) picture taken by the Gobabeb Research Center at the arrival of the flood mid-January 2021, and (**e**) the flood in the Kuiseb River after 4 days.

The climate over the Kuiseb catchment is dominated by a steep climatic gradient from the eastern highlands, with 500 mm/year of rainfall, to virtually no rainfall on the coast in

the West [3,17]. Along the lower Kuiseb River, there is too little rainfall, ranging from 10 to 50 mm per year, to sustain a river flow [16]. Episodic waterflows in the Kuiseb River are mostly due to rainfall that occurs in the upstream in the upper catchment on the highlands. Rainfalls in the Kuiseb catchment are predominantly due to summer thunderstorms from January to April, with 88% of floods occuring between December and April [15]. These floods have a duration from a few hours to a few days, which is why the Kuiseb River is called an "ephemeral river" [18]. The mean duration of Kuiseb floods is 12 days, and a "big" flood occurs, on average, every 25 years [15,18,19]. Flood events rarely reach the ocean as the water infiltrates, evaporates, or evapotranspirates before arriving in the deltaic zone. In the lower reaches, the riverbed forms the boundary between two distinct ecosystems: gravel plains to the north and the Namib Sand Sea to the south. Close to the ocean, this border is being forced towards the north over time by aeolian dune encroachment (Figure 1c, [20]). Over the last 20 years, there were only three large flood events in the Kuiseb River—in 2011, 2021, and in 2022. Photographs from the Gobabeb Namib Research Institute shows the arrival of a flood event at Gobabeb on 13 January 2021 (Figure 1d), and when it started to "disappear" on 17 January 2021 (Figure 1e).

Floods in the Kuiseb River are the main source of water to the coastal region of central Namibia, even if the water flow is rare, since flood water infiltration recharges the alluvial aquifers along the river [3,19]. Numerous alluvial aquifers occur along the Kuiseb River that not only provide essential water needs for humans, vegetation, and wildlife, but also for industry, tourism, and irrigation. These hidden treasures are essential for life during dry periods [21]. However, the recharge of alluvial aquifers will not be the same as it depends on the intensity and duration of a flood and the nature of the alluvial sediment.

The alluvium in the bed of the Kuiseb has a thickness of about 15–20 m with high permeability, as indicated by high pumping rates, which suggests rapid recharge of aquifers when flood events occur [21]. Indeed, at Gobabeb for example, recharge of the alluvial aquifer occurs at a rate of 0.5 to 1 cm/h when the water level in the river bed is greater than 15 cm. It was estimated that a flow of 18 days is necessary to fully recharge the aquifer at Gobabeb [18].

The hydrological regime of intermittent flooding of the Kuiseb River influences the growth and distribution of vegetation [22], resulting in lateral and longitudinal patterns of plant species occurrence and zonation due to their physiological characteristics, the availability of nutrients, and the water gradient of the aquifer. For example, the specie *Faidherbia albida*, which is one of the major trees along the course of the lower Kuiseb River, is very reliant on the recharge of aquifers and has the highest mortality rate during extended dry periods. Numbers of this species decline downstream as aquifer recharge becomes less regular [22].

However, the dynamics of the Kuiseb aquifers are still poorly known, but could be monitored using remote sensing data, as is shown in this study.

## 3. Data Used for This Study

### 3.1. Sentinel Missions

The Sentinel satellites constellation is composed of six satellites that has been developed by the European Space Agency (ESA) to support the Earth observations needs of the Global Monitoring for Environmental Security (GMES), which is a joint initiative between the European Commission and the ESA [23,24]. These satellites were designed to ensure data continuity from former missions, such as SPOT/Landsat for multispectral imagery and ERS-1-2 for radar, by providing daily observations of the Earth. A variety of remote sensing sensors on board the first three satellites, Sentinel-1, Sentinel-2, and Sentinel-3, provide long-term time-series of measurements with high spatial and temporal coverage [24]. Hence, data and products derived from these Sentinel missions offer a significant scientific opportunity [24] as sources of information to address global «grand challenges». For this study we used Sentinel-1 imaging radar and Sentinel-2 multispectral imaging sensor.

### 3.1.1. Multispectral Imagery from Sentinel-2

Multispectral images came from the Sentinel-2 mission, which is comprised two identical satellites, 2A and 2B, that were launched on June 2015 and March 2017, respectively, to continue SPOT (Satellite pour l'Observation de la Terre) for Earth observation [25,26]. The on-board MSI (Multi Spectral Imagery) sensors acquire data in 13 spectral bands, from visible to shortwave infrared (SWIR), with a spatial resolution from 10 to 60 m. Detailed characteristics of the Sentinel-2 mission are provided on https://sentinels.copernicus.eu/web/sentinel/missions/sentinel-2 (accessed on 10 September 2020). The two satellites were on the same level, but phased at 180°, to acquire images every 5 days from the same location. However, the quality of the images is strongly dependant on the weather.

Sentinel-2 Level 2A images are available for free download from https://peps.cnes.fr/rocket/#/home (accessed on 14 September 2020). Images were atmospherically corrected using the MAJA algorithm (MACCS ATCOR Joint Algorithm), which is a processing chain developed by the CNES (Centre National d'Etudes Spatiales) and CESBIO (Centre d'Etudes Spatiales de la Biosphère) to detect and apply atmospheric corrections [27]. This MAJA algorithm has shown good performance and accuracy in previous studies [28].

A total of 123 cloud-free images were acquired between June 2015 and December 2021, covering the Kuiseb River—and were analyzed. Spectral indexes were computed to monitor the vegetation and moisture variations through time. A spectral index corresponds to a ratio between two or more bands, which allows the standardization of data and the reduction of environmental and illumination effects. One of the most used spectral indexes is the NDVI (Normalised Difference Vegetation Index), which is defined by the following equation (Equation (1), [29]):

$$NDVI = \frac{NIR - Red}{NIR + Red} \tag{1}$$

with:

-   NIR band: central wavelength 842 nm (spectral band n°8)
-   Red band: central wavelength 665 nm (spectral band n°4)

In order to study water or moisture variations, we also considered the NDMI index (Normalised Difference Moisture Index), which is defined by the following equation (Equation (2), [30]):

$$NDMI = \frac{NIR - SWIR}{NIR + SWIR} \tag{2}$$

with:

-   NIR band: central wavelength 842 nm (spectral band n°8)
-   SWIR band: central wavelength 1610 nm (spectral band n°11)

However, for the purpose of our case, these two indexes were used to make a qualitative assessment to detect general patterns and variations. They could not be used to exactly quantify the vegetation and/or moisture contents.

### 3.1.2. Synthetic Aperture Radar (SAR) from Sentinel-1

Sentinel-1 is the first mission of the Copernicus Sentinel series, which is comprised of two identical satellites, 1A and 1B, in the same orbit, and were launched on April 2014 and April 2016, respectively. They provide data every 6 days from the same location in all weather conditions and during day or night. The Synthetic Aperture Radar (SAR) on board the Sentinel-1 mission is an active system that emits its own signal and then records the amount of energy backscattered after interactions with the Earth. A SAR is a coherent radar producing high-resolution images by emitting pulses lateral to the trajectory of the satellite in the centimeter wavelengths (band C~6 cm) and by measuring the backscattered energy of the objects. The main advantage, unlike optical data, is that data acquisition takes place during day and night, and even in overcast weather conditions—since clouds are transparent to the frequencies used. Due to the lateral scanning along the azimuth, the radar returns signals reflected by different objects on Earth that are detected at different distances

and are properly discriminated by measuring return times. A 2D image is then obtained according to the azimuth and the range direction using the synthetic aperture technique.

Radar imaging can provide polarimetric information for each pixel in the image, such as HH, VV, VH, and HV. For example, for VH polarization, the signal is transmitted in the vertical polarization and received in the horizontal polarization. The signal intensity from different polarizations yield information about the structure of the imaged surface, depending on the various types of scattering, such as the rough surface, bulk, and double bounce. Since the signal is acquired in a coherent way, the phase information therefore provides interferometric data. Each pixel X of a SAR image is composed of a real part and an imaginary part, which are related to the amplitude A and the phase $\varphi$ and are determined by the following equation (Equation (3), [31]):

$$X = A.\cos \varphi + j.\ A.\sin \varphi \tag{3}$$

When considering the information on the amplitude, we use the backscattering coefficient $\sigma_0$, which is defined as following (Equation (4)):

$$\sigma_0 = 10.\log A \quad \text{(in dB)} \tag{4}$$

As a reminder, the backscattering coefficient $\sigma_0$ can be used as a proxy for soil moisture variations through the impact of water on the dielectric properties.

Sentinel-1 carries a single C-band synthetic aperture radar instrument operating at 5.4 GHz in different polarizations, such as HH + HV, VV + VH, and VV and HH, and in four acquisition modes: StripMap (SM), Interferometric Wide Swath (IW), Extra-Wide swath (EW), and Wave Mode (WM). More information is available on https://sentinel.esa.int/web/sentinel/technical-guides/sentinel-1-sar (accessed on 5 January 2021).

Data on amplitude were obtained from Ground Range Detected (GRD) images and Single Look Complex (SLC) images for phase differences between two acquisitions—all in IW mode and in dual polarizations VV + VH—from tile Path 29 and Frame 1102, as StripMap mode and other polarizations were not available for our area of interest over Namibia. A total of 153 GRD images and 14 SLC images (28 interferometric couples), acquired October 2016 and December, were downloaded from https://search.asf.alaska.edu/#/ (accessed on 25 November 2021). GRD and SLC data were processed using the SNAP (SeNtinel Applications Platform) software, version 8.0.0. The different pre-processing steps are shown in Figure 2.

The various processing steps applied to the SLC, and therefore the phase (Figure 2), resulted in interferograms and coherence maps. An interferogram is obtained by the phase difference of two SLC images, a master image and a slave image, while coherence is a measure of the quality of the interferogram due to the complex correlation between a master image and its slave image. Coherence values are between 0 and 1, depending on the degree of correlation between the master and slave images, with a value of 1 implying perfect coherence between the two images and zero corresponding to a change in the state of the ground between the two acquisitions, thereby leading to a total loss of coherence.

### 3.2. Surface Backscattering Model

In order to understand the amplitude variations of the backscattering coeffcient $\sigma_o$ in SAR images, we used a surface backscattering model with a single layer. The "Integration Equation Model (IEM)" was developed by [32] and updated in subsequent years [33]. According to [20], the skin depth of the C-band radar signal for Sentinel-1, with a frequency of 5.4 GHz, is equal to 20 cm on a dry soil (dielectric constant $\xi' = 3.22$ and $\xi'' = 0.08$)—based on values measured in the field at Gobabeb [20]. As a first step, we therefore consider that the penetration of the radar signal is low, which justifies the use of a surface backscattering model. The IEM model simulates the backscattering coefficient $\sigma_0$ (in dB) of a surface from

the physical characteristics of the ground and for a given SAR configuration, depending on the following parameters [32,33]:

$$\sigma_0 = \text{function}(f, \theta, \xi, \sigma, L_{corr}) \tag{5}$$

with:

f = SAR frequency (GHz)
θ = SAR incidence angle
ξ = Dielectric constant (real part) of the surface
σ = Surface height standard deviation (cm)
$L_{corr}$ = Surface correlation length (cm)
(σ and $L_{corr}$ are the roughness parameters of the surface).

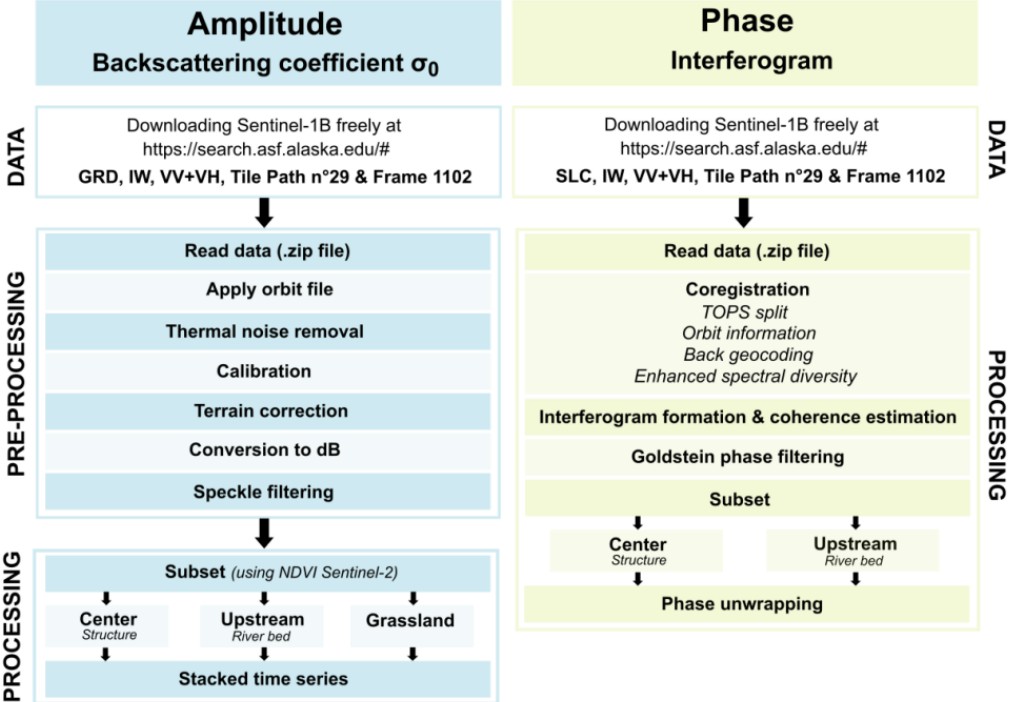

**Figure 2.** Flowchart of the processing steps for analysing both amplitude and phase of Sentinel-1 data. All the processing was done using SNAP (SeNtinel Applications Platform) software, version 8.0.0.

In our case, the frequency is equal to 5.4 GHz and ξ varies from 3 (dry environment) to 15 (wet environment). We therefore varied the σ and $L_{corr}$ parameters to find the best combination that reproduces $\sigma_0$ variations similar to those observed in the Sentinel-1 radar signal amplitude.

### 3.3. In Situ Data

#### 3.3.1. Meteorological Data

Meteorological data were obtained from the African Science Service Centre for Climate Change and Adaptive Land management (SASSCAL, http://www.sasscalweathernet.org/ (accessed on 17 March 2022). Although different parameters (air temperature, rain, fog, humidity, wind, etc.) were measured at hourly intervals, we only used rainfall data for Gobabeb (see Figure 1c for its location) and Claratal (located in the upper catchment of the Kuiseb River on the high plateau to the East).

#### 3.3.2. River Discharges

Daily river discharges measured at Gobabeb, Rooibank, and Schlesien, which were available until 2018, were downloaded freely from the GRDC (Global Runoff Data Cen-

tre) website (https://www.bafg.de/GRDC/EN/01_GRDC/grdc_node.html (accessed on 18 March 2022).

## 4. Results and Discussion

### *4.1. Spatial Variations of Vegetation and Water Spectral Indexes*

NDVI and NDMI were calculated from 123 relevant Sentinel-2 images between June 2015 and January 2022. A total of 18 images were acquired during the austral summer (December-January-February), 26 during autumn (March-April-May), 49 during winter (June-July-August), and 30 during spring (September-October-November).

Figure 3 depicts the average of the spectral indexes calculated over the time-series. Figure 3a,b shows the temporal average for the NDVI and the associated standard deviation, while Figure 3c,d provides the average for the NDMI and its associated standard deviation. Figure 3a indicates high NDVI values in the study area, particularly in the downstream, deltaic part of the lower Kuiseb River; in the center, near Klipneus; and in the upstream part, near Gobabeb. These three areas are delineated by black boxes in Figure 3a. The standard deviation (Figure 3b) also shows high values for these areas, indicating strong seasonal and interannual vegetation responses.

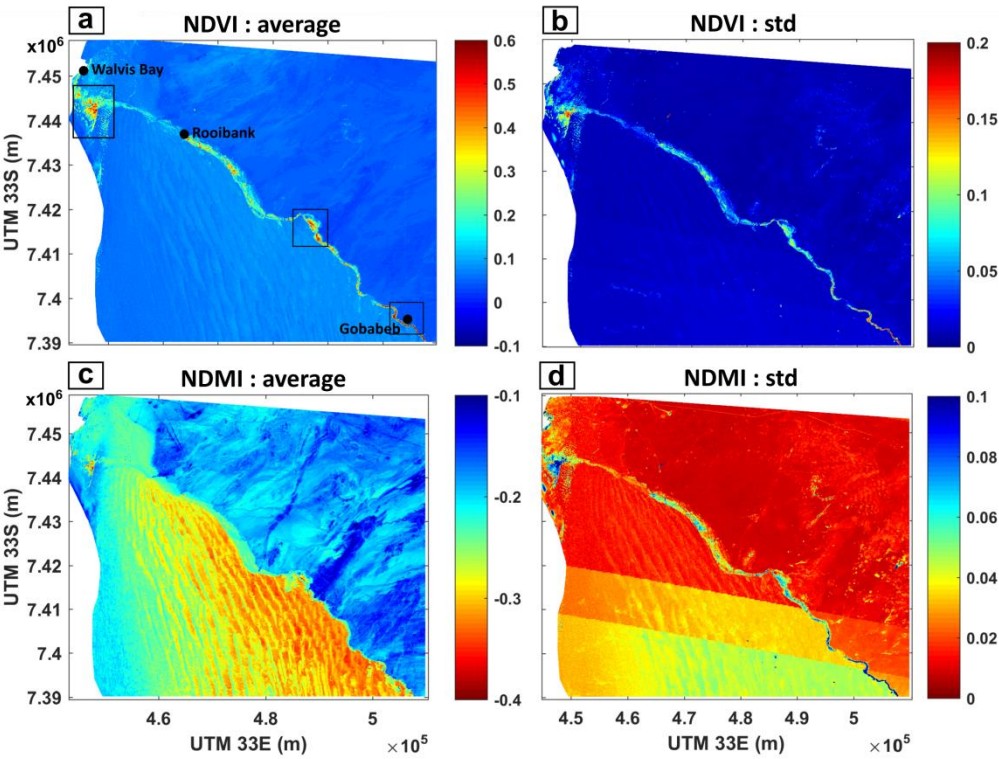

**Figure 3.** (**a**) Temporal averaged NDVI from Sentinel-2 images with names of major locations, (**b**) associated standard deviation a, (**c**) averaged NDMI from Sentinel-2 images, and (**d**) associated standard deviation. Black boxes in (**a**) show sub-areas selected for the future analysis. Main cities and villages are represented in black.

The averaged values of the NDMI (Figure 3c) are very different to the NDVI (Figure 3a), with high values north of the Kuiseb River on the gravel plains, and low values in the Namib Sand Sea to the south. Higher soil moisture in small channels and ephemeral tributaries of the gravel plains drainage to the Kuiseb River are clearly highlighted by higher average NDMI values, which are absent in the sand dunes. Even though low NDMI values south of the Kuiseb River—which correspond to the Aeolian dunes of the Namib Sand Sea—are not unexpected, a pronounced gradient of diminishing NDMI values from west to east indicate the effect of advective fog from the ocean on surface moisture in

windblown sand. Notably, Figure 3d indicates the high variations in soil moisture within the Kuiseb River at similar locations than the boxed areas delineated in Figure 3a.

These maps guided us to select and define three sub-areas with high NDVI and NDMI variations. On Figure 4a–c, we present NDVI maps for the box in the center of the lower Kuiseb study area (see Figure 3a). At Klipneus, a tributary with negative NDVI values appears on 5 May 2018 (Figure 4b, black box and red cross) and is no longer detectable on the next date (Figure 4c). Negative NDVI values indicate the presence of water or moisture in the Klipneus tributary that were only briefly present during May 2018. Meteorological data from nearby weather stations indicate a rain event at the time, thus the negative values are therefore linked to the presence of water on the surface.

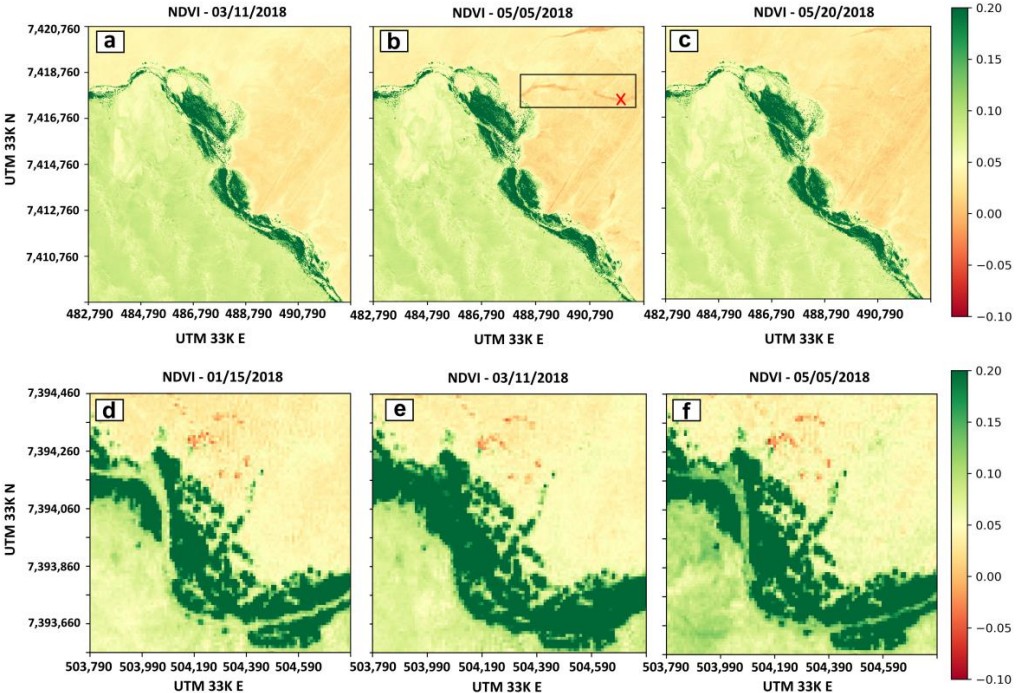

**Figure 4.** (**a**–**c**) NDVI maps in the central part (Klipneus) of the lower Kuiseb River acquired at three different dates (11 March 2018, 5 May 2018, and 20 May 2018) and (**d**–**f**) NDVI maps further upstream near Gobabeb acquired on 15th January 2018, 11th March 2018, and 5th May 2018. The black box in 4b represents the Klipneus tributary that were studied further.

Similar NDVI variations can also be observed in NDVI maps (Figure 4d–f) for the box located further upstream, near Gobabeb (see Figure 3a). On 11 March 2018 (Figure 4e), the NDVI index along the main course of the Kuiseb River increases, decreasing again by 5 May 2018 (Figure 4f). However, compared to Figure 4a–c, vegetation growth could not be linked to the rainfall or flood event during 9–12 March 2018; vegetation growth needs approximately one month after the rain or flood. The best explanation for that NDVI signal downstream of Gobabeb is microbial activity from flood debris (which occured before March 2018).

## 4.2. Temporal Variations of NDVI, NDMI and $\sigma_0$

After computing spectral indexes (NDVI, NDMI) for the 123 images of Sentinel-2, a time-series of the radar backscattering coefficient (notated as $\sigma_0$) for the same detailed areas were calculated from 149 Sentinel-1B scenes. Figure 5 shows time-series graphs of $\sigma_0$ in the VV polarization and spectral indexes at three locations in the study area: the Kuiseb delta downstream (Figure 5a,d,g), Gobabeb in the upstream part (Figure 5c,f,i), and at a tributary at Klipneus in the central part (Figure 5b,e,h). Graphs of rainfall at Gobabeb (Figure 5j,k,l)

and Claratal in the upper catchment of the Kuiseb River (Figure 5m,n,o) were repeated for each column of graphs.

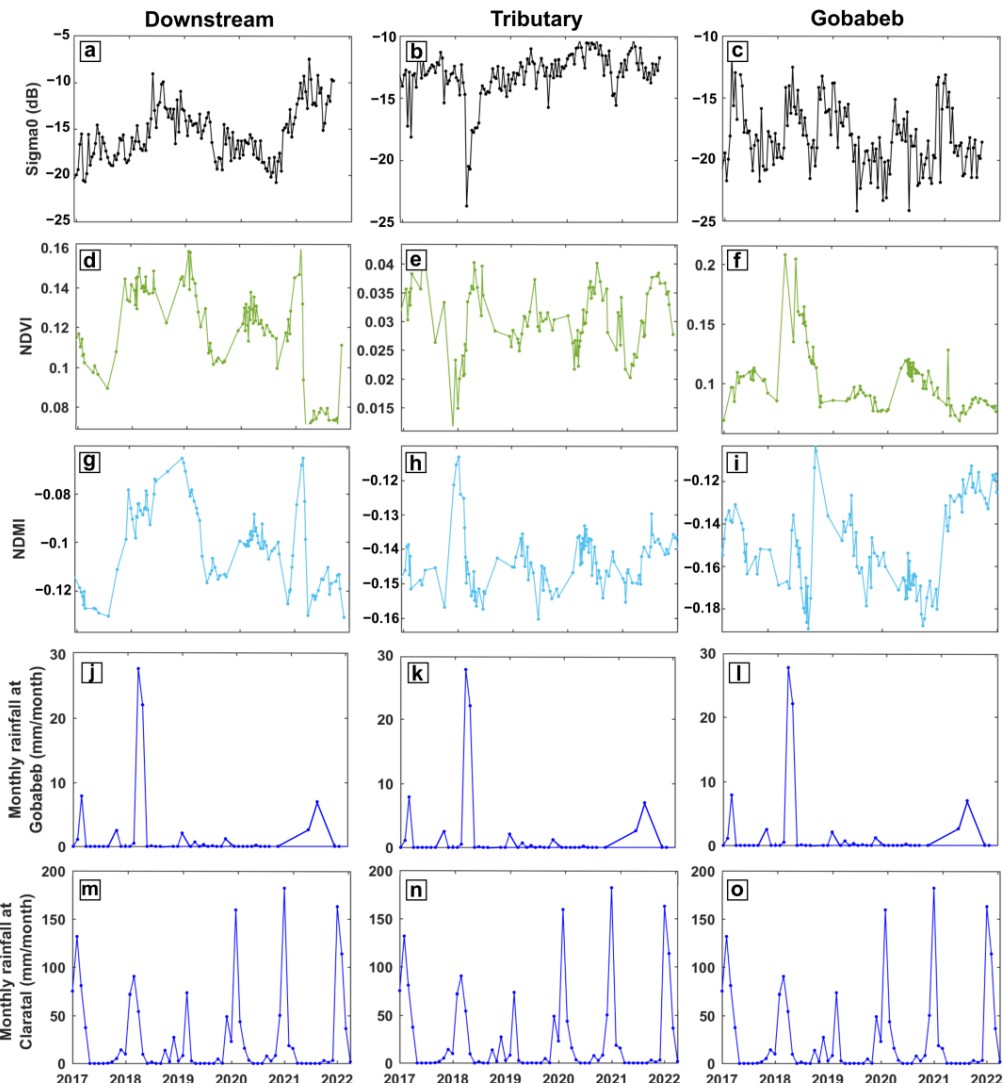

**Figure 5.** (**a**) Time-series of $\sigma_o$ (in VV polarization, in dB) at Gobabeb, (**b**) in the Klipneus tributary and (**c**) in the delta downstream. (**d**) Time-series of NDVI spectral index at Gobabeb, (**e**) in the Klipneus tributary and (**f**) in the delta downstream. (**g**) NDMI spectral index at Gobabeb, (**h**) in the Klipneus tributary and (**i**) in the delta downstream. (**j**–**l**) are time-series of rainfall (mm) at Gobabeb and (**m**–**o**) are time-series of rainfall (mm) at Claratal in the upper catchment of the Kuiseb River.

In the downstream deltaic area of the Kuiseb (Figure 5a), the amplitude of $\sigma_o$ shows important variations (around 7 dB) in 2018 and 2021. NDVI and NDMI spectral indexes (Figure 5d,g) depict similar temporal variations in 2018 and 2021. However, the peak of NDMI always precedes the NDVI peak. This delay corresponds to the time needed for vegetation growth after a moisture event. Not surprisingly, these variations are correlated with rainfall (Figure 5j,m), such as in 2018 and 2021. A series of pronounced flood events during January 2021 caused strong variations in NDMI, NDVI, and $\sigma_0$.

A tributary located at Klipneus in the center of the study site (Figure 5b) shows a considerable decrease of $\sigma_0$ (around 7 dB) in 2018, with an associated decrease in NDVI but an increase in NDMI (Figure 5e,h). The low NDVI values in April 2018 cannot be explained by in situ meteorological data from Gobabeb and Claratal (Figure 5k,n), as those sites recorded earlier rainfall events during March–April 2018—although the 2018 NDMI peak (Figure 5h) and rapid increase in NDVI during early 2018 (Figure 5e) are associated with

the rainfall recorded at Gobabeb. During the flood of January 2021, a small decrease of $\sigma_o$ was observed (Figure 5b), but variations were also visible in the NDMI and NDVI spectral indexes. As expected, the moisture index NDMI peak occurs before the vegetation index NDVI peak, with a shift of about 3–4 weeks (see 2018 for example), probably because even a slight increase in water supply results in rapid growth of vegetation in desert environments.

Upstream, at Gobabeb, variations in $\sigma_0$ (Figure 5c) correspond to seasonal variations with the highest peaks during March–April. Increases in NDMI were followed by higher NDVI (Figure 5f,i) within—approximately—a month after the event. In 2021, a high peak in all parameters was linked to the January 2021 floods and rain in the upper catchment, such as on Claratal (Figure 5o).

These results highlight the fact that even though vegetation and moisture variations are quite heterogenous along the lower Kuiseb River, these variations seem to be mainly linked to rainfall events. However, variations in $\sigma_0$ throughout the study site do not seem to be related to moisture variations. We therefore used the surface backscattering model developed by [33] to better understand $\sigma_0$ variations and to check if the variations in $\sigma_o$ in Sentinel-1 data, from $-22$ to $-17$ dB in the dry period versus $-17$ to $-12$ dB in the wet period, are related to only moisture variations without changes in roughness.

We attempted to reproduce the observed variations by setting the IEM model for a dry environment ($\xi = 3$), then varied parameter $\sigma$ (the surface height standard deviation) from 0 to 3 cm and $L_{corr}$ (the surface correlation length) from 0 to 10 cm. In the first simulation, only the impact of moisture, without roughness changes, was tested, but no value of $\sigma$ and $L_{corr}$ could explain the 5–7 dB variations in $\sigma_0$. Figure 6a shows an example with settings of $\sigma = 2$ cm and $L_{corr} = 0.25$ cm under dry ($\xi = 3$) and wet ($\xi = 15$) conditions—the difference between the two curves is less than 1 dB. Thus, moisture alone cannot explain the observed 5–7 dB variations in our $\sigma_0$ time-series of Sentinel-1 data. In a second simulation, we varied the input values for $\sigma$ and $L_{corr}$ in addition to moisture. Figure 6b shows how changes in the roughness parameter $L_{corr}$ resulted in a difference of 5–7 dB when switching from a dry to a wet environment. A change in roughness is also necessary together with the change in moisture for the expected result of 5–7 dB variation in $\sigma_0$ values. The change in roughness is probably due to the appearance of vegetation. Figure 6c shows the possible values of $\sigma$ and $L_{corr}$, outlined with black boxes, which produced a variation of 5–7 dB, wherein $\sigma$ must be less than 0.5 or greater than 2.3 cm and $L_{corr}$ must be between 0.1 and 0.8 cm. Thus, the variations we observed in the $\sigma_0$ time-series from Sentinel-1 data of the lower Kuiseb River are not only linked to variations in moisture or water availability, but also to changes in surface roughness, which were caused by vegetation growth.

*4.3. Variations at the Klipneus Kuiseb River Tributary*

We focused on a tributary at Klipneus (outlined in red in Figure 7a) to examine the hydrological dynamics in the central part of our study area (Figures 4b and 5b,e,h). The time-series of the backscatter coefficient $\sigma_0$ (Figure 7b) and the NDVI maps acquired in May 2018 (Figure 7c) and February 2021 (Figure 7d) show two negative peaks in 2018 and 2021 that are likely related to rain events (see part 4.2). The tributary is visible in multispectral imagery from 2018, but not in 2021, when an exceptional flood occurred in the Kuiseb River. We were interested in explaining these differences between the two years. Information on the properties of the near subsurface can be obtained from the backscattering coefficient $\sigma_0$, since the C-band penetration depth for Sentinel-1, with a frequency of 5.4 GHz, is in the order of approximately 20 cm for dry soils. In comparison, the MultiSpectral Imagery sensor on Sentinel-2 only allows for measurements from the surface. We also obtained additional information from the phase of the Sentinel-1 radar signal, which is more sensitive to moisture variations.

Meteorological records from in situ stations in the Kuiseb River catchment recorded rain in 2017 and 2018 in the lower Kuiseb (Figure 5k), including at Klipneus (*pers. obs.* Marais), which explains the variations in Sentinel-1 amplitude and Sentinel-2 NDVI vegetation growth for those years. However, in 2021, after a large flood caused by heavy

rains upstream in the catchment of the Kuiseb River (Figure 5n), no change in the NDVI was visible at the tributary location, yet there was a visible decrease in the $\sigma_0$ amplitude (Figure 7b).

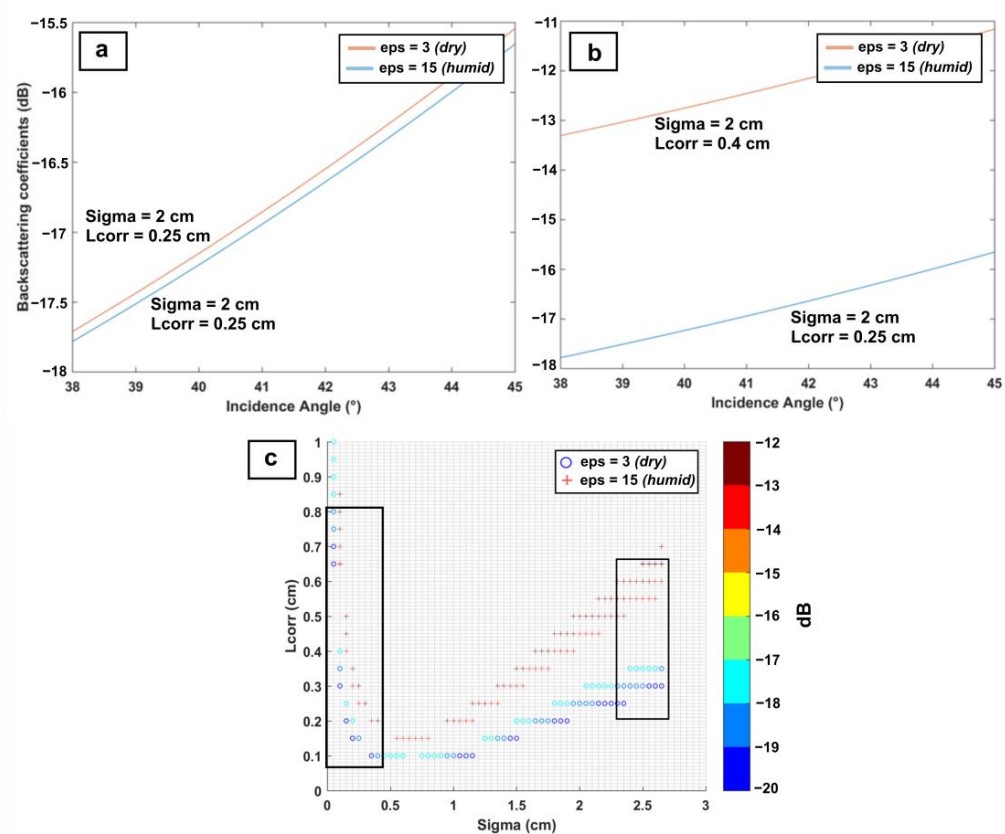

**Figure 6.** Surface backscattering model outputs with (**a**) fixed $\sigma$, Lcorr in dry/humid conditions; (**b**) different $\sigma$, $L_{corr}$ in dry/humid conditions; and (**c**) $\sigma$, $L_{corr}$ values in the black boxes allow us to reproduce $\sigma_0$ variations observed in the Sentinel-1 time-series from the lower Kuiseb River.

Radar interferograms at the tributary location were produced for 2018 and 2021 to try to understand the origin of the variations in the radar backscattering coefficient (Figure 7b). Interferograms mapped coherences in radar returns between two acquisition dates for 14 and 26 March 2018 (Figure 8a), 26 March and 7 April (Figure 8b), 25 May and 6 June (Figure 8c), and 18 June and 30 June (Figure 8d). Figure 8a (14 and 26 March 2018) corresponds to the state before the « disturbance », with Figure 8b,c during the « disturbance », and Figure 8d after the « disturbance ». « Disturbance » is defined by a strong decrease in coherence. It is observed over the entire area in Figure 8b,c, but especially for the tributary (indicated by a black square in Figure 8b). This decrease in coherence can be explained by 5 days of rain that occurred between March and June 2018. Interferograms were also produced for 2021, when a major flood occurred during January in the main channel of the Kuiseb River (Figure 8e–h). As before, the coherence maps correspond to different states before the flood (Figure 8e, 4 and 20 December 2020), during the flood (Figure 8f, 9 and 21 January 2021), and after the flood (Figure 8g, 22 March and 3 April 2021, and Figure 8h, 5 and 17 November 2021). Unlike 2018, there is no overall decrease in coherence over the entire area, but a localized decrease is visible in the tributary (indicated by a black square in Figure 8f). This decrease in coherence seems to be associated with the flood event in the Kuiseb river channel during January 2021.

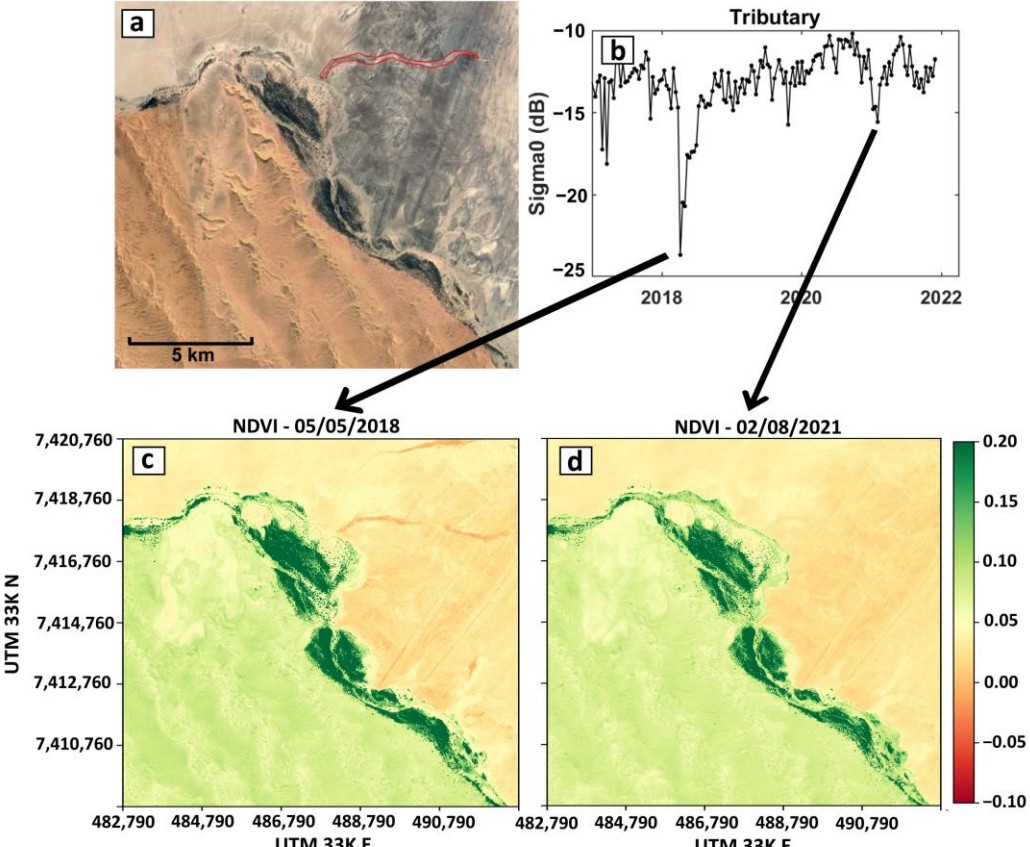

**Figure 7.** (**a**) The tributary at Klipneus in the central part of our lower Kuiseb River study area, (**b**) changes of the amplitude $\sigma_0$ (VV polarization) over the 2016–2022 period in the tributary, (**c**) corresponding NDVI map from 2018, and (**d**) NDVI map from 2021.

Hydrological activity in the Klipneus tributary seems to be linked to local rainfall, such as the 2018 event, which also affected a wider area as an interferogram « disturbance » (Figure 8b), or as a more localized « disturbance » that is linked to water flow in the main channel of the Kuiseb (Figure 8f). The large flood in 2021, which lasted 7 days, recharged aquifers and raised underground water tables along the Kuiseb River and in its floodplains. The interferogram (Figure 8f) shows that it is likely to have increased the water level in the Klipneus tributary too.

A possible explanation of the process is illustrated in Figure 9. In Figure 9a the tributary is outlined in red, with the Kuiseb floodplains in green and water flow along the main channel in blue. Figure 9b,c are photographs of a flood event. Figure 9d represents an idealised topographic profile along a transect in Figure 9a (in white) across the Kuiseb River bed and the Klipneus tributary. During a flood event, water will infiltrate into the river bed (at a rate of 8–12 mm/h, after [3]) and from overspill on the floodplain, but some water will be lost through evapotranspiration, evaporation, and pumping water for human activities. After a flood event, as presented in Figure 9e, groundwater levels will rise due to the recharge from infiltration, which would be the origin of variations in returns from the radar signal (coherence) in the tributary. As moisture levels rise and come closer to the surface, it will produce a change in the interferometric phase, i.e., a drop in coherence between different dates of acquisition. Groundwater level rise is the cause of an increase in surface moisture, resulting in a decrease in the coherence measured by the Sentinel-1 data. Indeed, the phase depends on the dielectric properties of the soil and will be sensitive to the presence of water/moisture.

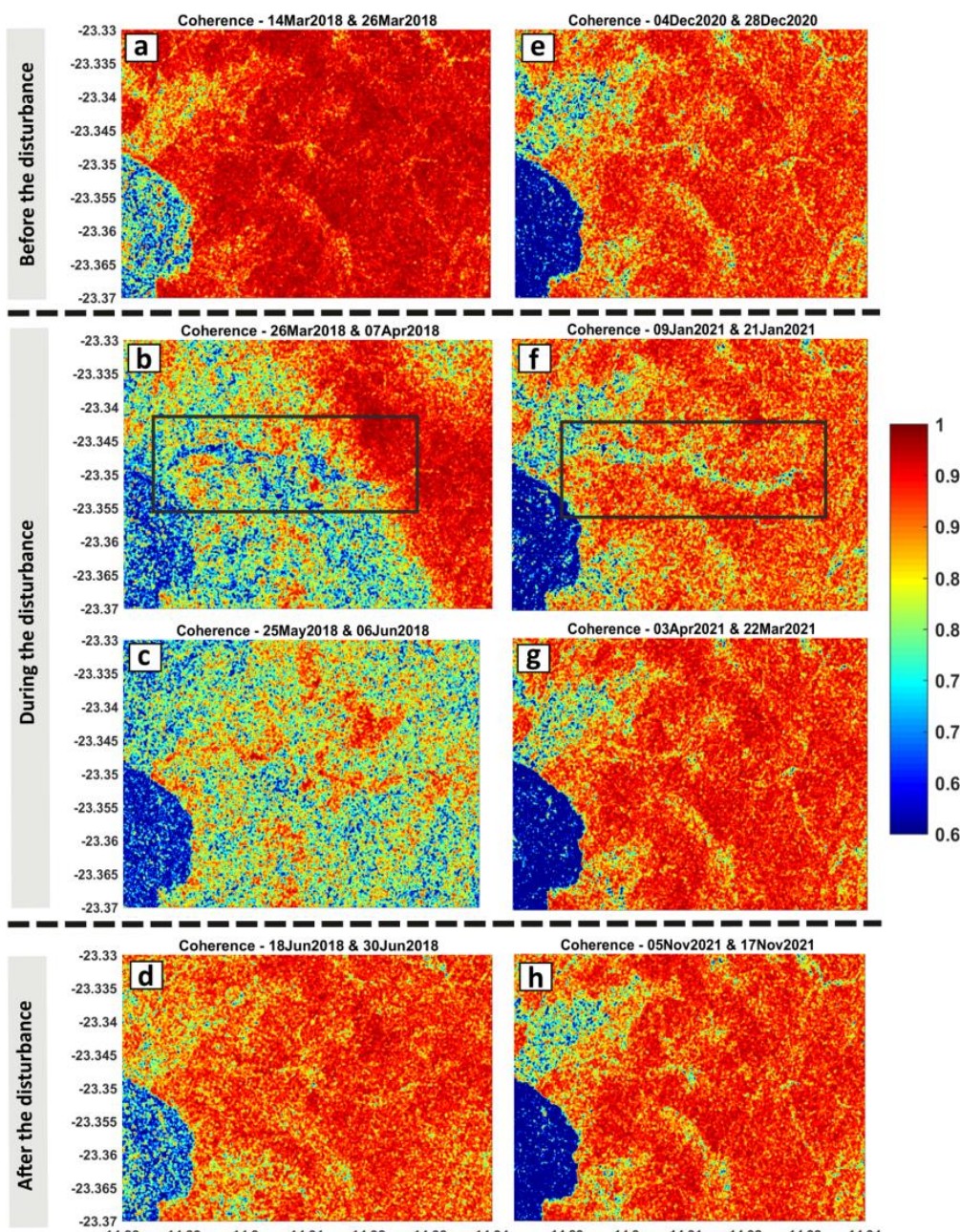

**Figure 8.** Coherence maps associated with the tributary area at Klipneus in Figure 7a, adjacent to the Kuiseb river channel. (**a–d**) are coherence maps associated with a local rain event in 2018; (**e–h**) are coherence maps associated with a flood event in 2021.

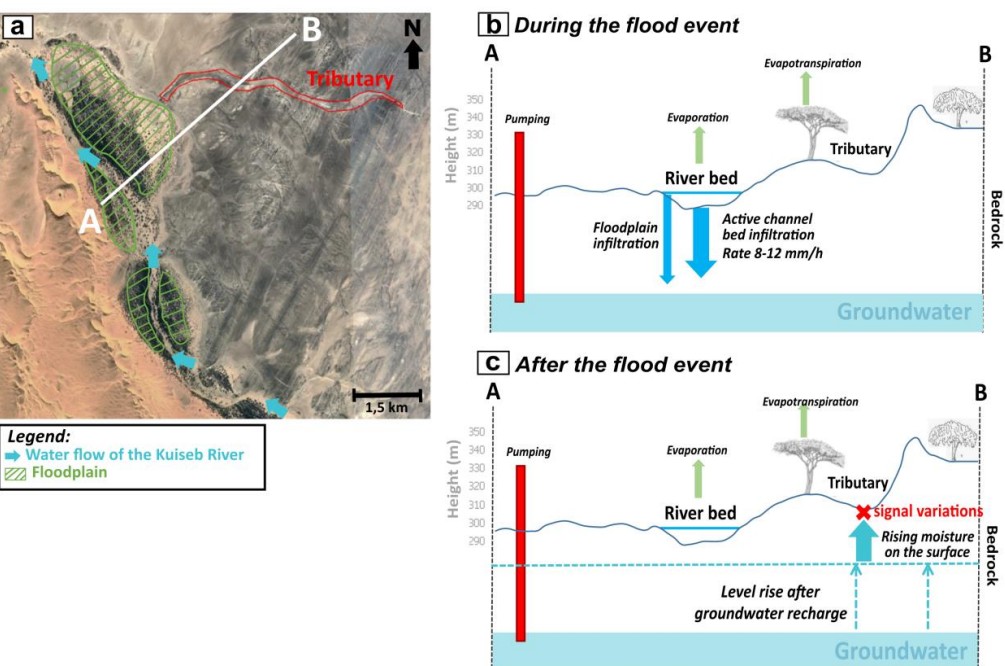

**Figure 9.** (**a**) Google Earth image showing the tributary outlined in red, floodplains outlined in green, and the direction of the water flow in the Kuiseb River during a flood event. The white line is a transect across the Kuiseb River and its tributary at Klipneus. (**b**) Idealised topographic section along a transect (white line in a) to illustrate likely hydrological consequences during and (**c**) after a flood event in the main river channel and the tributary that leads to a reduction in radar image coherence.

## 5. Conclusions

The Kuiseb River is a major ephemeral river in Namibia with episodic flood events that typically only last a few days. Until recently, studying the dynamics of such ephemeral rivers was difficult, mainly due to inappropriate monitoring tools as in situ hydrological and geohydrological stations are rare in episodically-flowing rivers of arid and semi-arid areas. However, the development of remotesensing data over the last two decades has offered new tools for the study of hydrological phenomena in ephemeral rivers. The main objective of our study was to show the potential of using Earth Observation Satellite remote-sensing data to monitor the dynamics of the Kuiseb River, using an innovative approach based on time-series of data from multiple sensors on different platforms provided by the Sentinel missions.

First, Sentinel-2 data were used to produce NDVI and NDMI maps in order to detect changes and define sub-areas for closer study. Local rainfall and flood events in 2018 and 2021 impacted these two spectral indexes, with an offset shift of one month between them, which corresponds to the time for vegetation to grow. Then, Sentinel-1 data were used to produce $\sigma_0$, which showed variations during local rainfall events in 2018, but also following a big flood event in 2021. Finally, coherence maps were produced for 2018 and 2021, which showed a decrease in the signal from a shallow tributary in 2018 and 2021. These observations can be explained by a rise in groundwater levels after the Kuiseb River floods of 2021. This study shows the utility of combining multi-sensor satellite data to better understand the actual hydrological dynamics of episodically flowing drainage systems, such as the Kuiseb River, following rainfall and flood events—and particularly their effects on groundwater levels and local aquifers.

This study was a preliminary work to show the feasibility of Sentinel data to better understand the dynamics of the ephemeral Kuiseb River by establishing a link between underground and surface dynamics. The different methods deployed during this work could be used in the future by institutions, for example, to automate all of these methods

to create a temporal monitoring tool from Sentinel-1 and Sentinel-2 data for different ephemeral rivers.

All the results produced during this study can be used by water resource managers in desert areas. The different maps from Sentinel-1 and Sentinel-2 can be used to locate groundwater risings, which are useful for the local population, but also animals. In addition, this study will allow them to better understand the link between the flow of surface and underground water, thereby monitoring the ephemeral river.

**Author Contributions:** Conceptualization, C.N. and P.P.; methodology, C.N. and P.P.; writing—original draft preparation, C.N. and P.P.; writing—review and editing, P.P., S.L., E.M. and K.S. K.S. was responsible for Living Planet Fellowship supervision of C.N. E.M. helped to provide meteorological data and hydrological data. All authors have read and agreed to the published version of the manuscript.

**Funding:** This research was funded by European Space Agency in the framework of a Living Planet Fellowship.

**Conflicts of Interest:** The authors declare no conflict of interest.

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
