# Peer review of "Monitoring the Dynamics of Ephemeral Rivers from Space: An Example of the Kuiseb River in Namibia"

_water, doi:10.3390/w14193142_

Round 1

Reviewer 1 Report

The manuscript water 1901510 presents a case study employing remote sensing techniques to study the hydrodynamics of an ephemeral river in Namibia.  The manuscript is well written and the story is well laid out.  I would suggest that the authors add a paragraph at the end of the article to inform the reader how this might be used by water resource managers in this area.  I also attached a file with a few minor comments. 

Author Response

Dear Editor and reviewer,

Please see the attachement for the reply.

Kind regards,

Reviewer 2 Report

see attached

Author Response

(The authors gave the same response as above.)

Round 2

Reviewer 2 Report

The authors answered my concerns sufficiently. It would be nice to have a study with more correlation and analysis against ground data, but that is not available for this region.